# Oxidative Stress in Amyotrophic Lateral Sclerosis: Synergy of Genetic and Environmental Factors

**DOI:** 10.3390/ijms23169339

**Published:** 2022-08-19

**Authors:** Anca Motataianu, Georgiana Serban, Laura Barcutean, Rodica Balasa

**Affiliations:** 1Department of Neurology, “George Emil Palade” University of Medicine, Pharmacy, Science and Technology of Targu Mures, 540136 Targu Mures, Romania; 21st Neurology Clinic, Emergency Clinical County Hospital Targu Mures, 540136 Targu Mures, Romania; 3Doctoral School, “George Emil Palade” University of Medicine, Pharmacy, Science, and Technology of Targu Mures, 540142 Targu Mures, Romania

**Keywords:** amyotrophic lateral sclerosis, oxidative stress, genetic factors, environmental factors, neurodegeneration

## Abstract

Amyotrophic lateral sclerosis (ALS) is a grievous neurodegenerative disease whose survival is limited to only a few years. In spite of intensive research to discover the underlying mechanisms, the results are fairly inconclusive. Multiple hypotheses have been regarded, including genetic, molecular, and cellular processes. Notably, oxidative stress has been demonstrated to play a crucial role in ALS pathogenesis. In addition to already recognized and exhaustively studied genetic mutations involved in oxidative stress production, exposure to various environmental factors (e.g., electromagnetic fields, solvents, pesticides, heavy metals) has been suggested to enhance oxidative damage. This review aims to describe the main processes influenced by the most frequent genetic mutations and environmental factors concurring in oxidative stress occurrence in ALS and the potential therapeutic molecules capable of diminishing the ALS related pro-oxidative status.

## 1. Introduction

Amyotrophic lateral sclerosis (ALS) etiology is not yet completely understood despite extensive research. Roughly 10% of ALS cases belong to a familial, mainly autosomal dominant inheritance pattern, while the remaining 90% are sporadic forms with no apparent genetic basis [1]. Numerous external occupational and environmental factors have been associated with ALS, including exposure to different chemicals, metals, and pesticides, electromagnetic fields (EMFs), and lifestyle choices, such as smoking and excessive physical exercise [2]. Nevertheless, these factors do not directly cause ALS but act upon various internal susceptibility factors and lead to ALS development. Over 30 mutations have been found to correlate with ALS, particularly its familial form, in genes, such as superoxide dismutase 1 (SOD1), transactive response (TAR)-DNA binding protein (TARDBP, previously called TDP43), angiogenin (ANG), fused in sarcoma RNA binding protein (FUS), and chromosome 9 open reading frame 72 (C9orf72) [3,4]. ALS is a multifactorial disease caused by various defective cellular and molecular processes, including glutamatergic excitotoxicity, axonal transport, RNA and protein metabolism, mitochondrial dysfunction, and oxidative stress (OS) [3,5,6].

Mitochondria are organelles of great importance in the human body due to their role in converting stored energy into adenosine triphosphate (ATP) via oxidative phosphorylation and phospholipid biogenesis, calcium buffering and regulating programmed cell death [1]. Oxidative phosphorylation is the primary source of free radicals, such as reactive oxygen (ROS), nitrogen (RNS), and sulphur (RSS) species, which are found in all cells but in limited amounts due to the counteracting antioxidant mechanisms [7].

The brain has the greatest energy demand relative to its size. Therefore, ATP production is essential for good neuronal metabolism, growth, and plasticity [1]. Consequently, neurons are the most susceptible to injury in mitochondrial dysfunction [8]. Mitochondrial damage and OS, in particular, are important in the complex ALS pathogenesis. However, their roles are not yet fully understood. This review highlights the principal pathways through which OS caused by genetic and environmental factors leads to motor-neuron degeneration, in addition to several ALS therapeutic alternatives related to OS.

## 2. OS and Mitochondrial Dysfunction in ALS

OS arises due to disequilibrium caused by excessive ROS production and insufficient compensatory antioxidant systems. The oxygen-free radicals are oxygen species produced by incomplete oxygen reduction in different enzymatic and non-enzymatic cellular processes [9,10]. Mitochondria are the OS mechanism’s primary target and the largest ROS producer [11].

Mitochondria are the primary source of ROS due to their role in ATP production via oxidative phosphorylation, whose significant adverse effect is the production of unpaired electrons [12]. The electron transport chain comprises five multiprotein complexes that mediate the interaction between these electrons and oxygen, creating ROS such as hydrogen peroxide (H_2_O_2_), superoxide anions (O2^−^), and hydroxyl radicals (HO^−^) [10,13]. Mitochondrial complex I (reduced nicotinamide adenine dinucleotide [NADH] coenzyme Q reductase) catalyses the electron transfer from NADH to ubiquinone (coenzyme Q). Ubiquinone also receives electrons from complex II (succinate dehydrogenase). The reduced ubiquinone donates its electrons to complex III (cytochrome bc1) and, eventually, to cytochrome c (cytC). Complex IV (cytC oxidase) participates in the interaction between molecular oxygen and the electrons removed from cytC, leading to water formation [1]. Complexes I, II, and III are the most frequently associated with premature electron leakage to oxygen and play a significant role in ROS production [14].

In addition, increased ROS levels lead to the formation of other reactive species, such as RNS, due to O2^−^ interacting with other molecules, such as nitric oxide, to form peroxynitrite (ONOOˉ). Moreover, in addition to ROS and RNS, mitochondria produce RSS, which are also incredibly reactive. Free oxygen radicals progressively damage proteins, lipids, and nucleic acids, resulting in inefficient cellular processes, inflammation, and cell death [1,3]. Mitochondrial internal components and mitochondrial DNA (mtDNA), in particular, are highly susceptible to OS-induced damage, eventually hindering normal mitochondrial bioenergetics, increasing ROS production and OS [15]. The protective antioxidant systems comprise both enzymatic and non-enzymatic processes. The key enzymes involved in catalytic ROS removal are superoxide dismutases (SODs), catalase (CAT), glutathione peroxidases (GPXs), glutathione reductase (GR), and thioredoxin (TRX). In addition, the non-enzymatic complexes primarily comprise vitamins A, C, and E, glutathione (GSH), and proteins such as albumin and ceruloplasmin [10,16].

Abnormally high free radical levels and low antioxidant systems represent a universally accepted yet incompletely understood pathological ALS characteristic. OS unquestionably plays a crucial role in motor neuron death, but the precise timing of oxidative damage remains unknown [17]. OS biomarkers have been identified in ALS patients’ brain tissue, cerebrospinal fluid (CSF), blood, and urine [18]. Because the life expectancy of ALS patients is relatively short, it is impossible to monitor OS biomarkers over an extended period. Moreover, it is challenging to determine whether OS is a cause of ALS-associated neurodegeneration or a consequence of other underlying etiologic factors because of its sporadic onset and the current lack of methods to predict ALS development [9]. Studies with a murine ALS model have found modified mitochondrial structures and nuclear factor erythroid 2-related factor 2 (Nrf2) pathway activation, which generally occurs due to OS-induced damage and stimulates the formation of intracellular antioxidant molecules, during early ALS stages, implying OS involvement in the initial phase of ALS [19,20].

However, these studies used the murine mutant SOD1 ALS model, and SOD1 mutations only account for 20% of human familial ALS cases. Babu et al. [21] found a significant increase in lipid peroxidation and decreases in antioxidant enzymes CAT, GR, GSH, and glucose-6-phosphate dehydrogenase (G6PD) in the erythrocytes of 20 sporadic ALS patients. The changes progressed alongside ALS pathogenesis, consistent with OS involvement in ALS development. Furthermore, the abovementioned environmental and occupational risk factors mutually stimulate and induce pro-oxidative states, which can eventually adversely affect motor neurons [22].

Coiled-coil-helix-coiled-coil-helix domain-containing protein 10 (CHCHD10) is a mitochondrial protein located in the intermembrane space (IMS) with no recognised function [23]. However, it is believed to be involved in maintaining mitochondrial cristae morphology and proper oxidative phosphorylation. Overexpression of mutant CHCHD10 containing an allele associated with ALS leads to altered mitochondrial structure and defective electron transport chain activity, particularly in multiprotein complexes I, II, III, and IV [24,25]. Moreover, fibroblasts had mitochondrial ultrastructural damage and mitochondrial network fragmentation in an ALS patient carrying a CHCHD10 mutation [24].

## 3. Genetic Variants and OS

### 3.1. SOD1 Mutations

SOD1 is located on chromosome 21 and encodes an important intracellular antioxidant enzyme. SOD1 is found mainly in the cytosol, with approximately 5% of total cellular SOD1 found in the mitochondrial IMS. It primarily converts the O2^−^ into H2O2, which is then converted to water and oxygen by other antioxidant enzymes, such as CAT, GPXs, and peroxiredoxin [26]. An alternative process for removing primary ROS is detoxification by cytC, the main advantage of which is the absence of secondary ROS. However, its major drawback is its interaction with H2O2 resulting from SOD1 activity (cytochrome peroxidation), leading to a highly reactive molecule, oxoferryl-cytC [26].

To adequately fulfil its function, SOD1 must reach a mature, highly stable form via complex post-translational modifications (PTMs) facilitated by a copper chaperone for SOD1 (CCS) [27]. PTMs include zinc and copper metal binding, disulphide bond formation and folding, and exposure of hydrophobic regions for further dimerisation [1]. Cytosolic SOD1 can traverse the outer mitochondrial membrane (OMM) through the translocase of outer membrane (TOM). Its return is prevented by CCS-mediated establishing disulphide bonds and inserting metal ions [1]. CCS distribution influences SOD1 localisation. Cytosolic CCS impedes SOD1 mitochondrial import, while mitochondrial CCS prevents SOD1 cytosolic export. Moreover, it acts as an oxygen sensor. Hyperoxia maintains cytosolic CCS, while hypoxia promotes the mitochondrial import of CCS and SOD1 [27]. Therefore, increased mitochondrial respiratory chain activity, CCS intra-mitochondrial translocation, and SOD1 maturation under hypoxia act as a compensatory antioxidant system [3]. In addition, the Mia40/Erv1 pathway participates in CCS mitochondrial import in a respiratory chain-dependent manner [27].

SOD1 has been implicated in ALS pathogenesis, with more than 160 mutations so far identified that typically change only a single amino acid. However, the exact mechanism leading to motor neuron death remains to be determined. While SOD1 mutations are primarily found in familial ALS cases, they likely also contribute to sporadic ALS [28]. Notably, mitochondrial accumulation of mutant SOD1 protein is characterised by increased enzymatic function, which appears to cause neurodegeneration [22,28,29,30]. However, the precise mechanisms of SOD1 toxicity are not fully understood. Nevertheless, several hypotheses have been proposed. SOD1 has a low level of intrinsic peroxidase activity, which can be amplified by high ROS levels [31]. SOD1 containing the Av5, H48Q, and G93A mutations has enhanced peroxidase activity and catalyses H2O2 conversion to HOˉ, irreversibly inactivating the dismutase structure [31]. Furthermore, O2^−^ shows a higher propensity for nitric oxide than mutant SOD1, leading to ONOOˉ production with further tyrosine nitration of cellular proteins, resulting in neuronal death [9,32]. SOD1 maturation is a complex process and is highly susceptible to disruption. Numerous amino acid mutations affecting metal binding or disulphide bridge formation lead to misfolded proteins, each of whose structure is unstable and inclined to create insoluble SOD1 aggregates. However, mutant SOD1 cannot adequately respond to CCS-induced PTMs, eluding the maturation process essential for normal function and enhancing intracellular ROS accumulation [33]. A negative feedback cycle occurs in which OS and mitochondrial damage caused by misfolded SOD1 lead to further SOD1 misfolding and further mitochondrial damage [1]. However, mutant SOD1 aggregates interact with OMM proteins involved in mitochondrial apoptosis, such as Bcl12 and voltage-dependent anion channel (VDAC), activating pro-apoptotic pathways. Regardless of the mechanisms involved, motor neuron death results [3,34] (Figure 1).

### 3.2. TARDBP Mutation

TARDBP, also known as TDP-43, is located on chromosome 1 and encodes a ubiquitous heterogeneous nuclear ribonucleoprotein comprising four domains: two RNA highly conserved recognition motifs essential for its role in protein biogenesis, a C-terminal glycine-enriched low complexity domain (LCD) involved in protein-protein interactions, and an N-terminal region whose function remains contentious [35,36]. It has various functions, particularly in RNA transcription, maturation, transport and translation, and plays a crucial role in intracellular stress management. TARDBP participates in the biogenesis and maintenance of stress granules, small membrane-less structures that form due to cellular exposure to stress and protect RNA and its associated ribonucleoproteins [37]. Moreover, TARDBP interacts with numerous proteins involved in diverse physiological processes, such as RNA metabolism, the immune response, and stress-induced pathways [38].

Almost 10% of ALS familial cases are associated with TARDBP mutations, which mostly affect its LCD. The key properties of mutated TARDBP include an increased tendency to aggregate, cytoplasmic mislocalisation, unstable structure, protease resistance, and altered protein-protein interactions [39,40]. Furthermore, patients with sporadic ALS frequently have elevated TARDBP levels within neuronal and cytoplasmic inclusions, a current hallmark of ALS pathogenesis [40]. Altered cellular redox balance has also been suggested as a causal factor in ALS pathogenesis. Several studies have demonstrated a reciprocal relationship between OS and TARDBP mutations: whereas TDP-43 mutations impair mitochondria, OS production greatly increases TDP-43 toxicity [41,42]. On the one hand, Cohen et al. [43] showed that OS induces cysteine disulphide cross-linking in TARDBP, decreasing protein solubility and enhancing the formation of insoluble cytoplasmic aggregates. More recently, OS was found to promote the acetylation of lysine-145, particularly in cytoplasmic TARDBP, leading to its aggregation, LCD hyperphosphorylation, and loss of normal TARDBP function [44]. On the other hand, Magrane et al. [45] have shown that mutated TARDBP (A315T) overexpression impairs mitochondrial structure and transport. Conversely, Wang [46] showed that both wildtype and mutated TARDBP (Q331K and M337V) reduced mitochondrial density in neuritis and mitochondrial dynamics. Altered mitochondrial function can enhance TARDBP aggregation through reduced GSH levels, leading to unbalanced ROS overproduction and neurodegeneration. Moreover, several studies have found that mutated TARDBP markedly decreased the antioxidant expression of Nrf2, further enhancing OS [43,44,47,48] (Figure 1).

### 3.3. FUS Mutation

FUS is located on chromosome 16 and encodes a member of the RNA/DNA-binding protein family that comprises two main domains: an N-terminal LCD region involved in transcriptional activation and a C-terminal region implicated in RNA and protein binding [49]. Similar to TARDBP, most mutations occur within the last 12 amino acids of the C-terminal region, particularly in familial ALS forms. However, N-terminal domain mutations are often associated with sporadic ALS [50]. Patients with FUS mutations typically develop juvenile ALS, with onset before the age of 40 years [49]. Due to its nucleic acid binding capacity, FUS is implicated in RNA metabolism and DNA repair [51].

FUS and TARDBP share similar molecular mechanisms that lead to ALS occurrence, both involving the cytosolic accumulation of protein aggregates that are a hallmark of ALS [50]. Nevertheless, FUS has been found to cause ALS by a DNA-related process. Due to their high metabolic demands, neurons are characterised by intense energy production, with a detrimental elevation in ROS, which is harmful to DNA. Wang et al. [51] showed that FUS mutants, particularly R521H and P525L, failed to mend OS-induced DNA damage properly. Consequently, significant numbers of unrepaired DNA strand breaks accumulate in the neurons, eventually leading to motor neuron death (Figure 1).

### 3.4. C9orf72 Mutation Associated with OS

C9orf72 is the most commonly mutated gene in both familial and sporadic ALS, located on chromosome 9 and is responsabile for RNA accumulation in the nucleus. It comprises a hexanucleotide sequence GGGGCC (G4C2) that is repeated at least 30 times within the gene’s first intron [52]. These repeats result in the production of abnormally structured and aggregated proteins, also known as five dipeptide repeat proteins (DPR), such as poly-glycine-alanine (poly-GA), poly-glycine-proline (poly-GP), and poly-glycine-arginine (poly-GR) [9,53]. Mori et al. [51] found that cytoplasmic inclusions, a hallmark of ALS, contain DPRs, particularly poly-GA. Several studies have shown that C9orf72 mutation contributes to motor neuron death by different gain- and loss-of-function mechanisms that disrupt normal glial, neuronal, and immune functions [52,54]. Another underlying neurodegeneration process is mitochondrial damage caused by pathologic protein-protein interactions between poly-PRs and mitochondrial ribosomal proteins. Onesto et al. [55] found hyperpolarisation of mitochondrial membranes due to increased ROS production in mutant C9orf72 fibroblasts derived from ALS patients’ blood. The resulting OS damages DNA and impairs its adequate repair in an age-dependent manner [56]. Furthermore, Birger et al. [57] confirmed OS involvement in ALS pathogenesis associated with mutant C9orf72, where astrocytes carrying the C9orf72 mutation inhibited the production of antioxidant compounds, enhancing OS not only in themselves but also in wildtype motor neurons (Figure 1).

### 3.5. Other Mutations Associated with OS

ANG is a ribonuclease that cleaves transfer RNA (tRNA), inhibiting translation initiation and enhancing stress granule formation [58]. It also activates the Nrf2 pathway in astrocytes, an intracellular mechanism protecting adjacent neurons against OS [59]. However, ANG is frequently mutated in ALS patients, compromising its neuroprotective function [60].

Paraoxonases (PONs) are a family of three distinct enzymes, PON1, PON2, and PON3, encoded by genes located on chromosome 7 [61]. PONs have potential antioxidant functions as they prevent lipid oxidation and neutralise toxic organophosphates (OPs) [9,22,61]. However, their importance in ALS remains contentious. While Ticozzi et al. [62] identified at least seven mutations in PON genes associated with familial and sporadic ALS, a comprehensive meta-analysis and case-control studies have found no significant correlations [63,64].

In the last decade, genomic studies using exome sequencing technologies have identified other genes involved in ALS pathogenesis, which are implicated in various molecular mechanisms, including OS production. Nevertheless, while their precise functions remain largely unknown, they might prove valuable for developing a personalised ALS approach [65].

## 4. Environmental Factors Associated with OS in ALS

Already recognised and exhaustively studied mutations can no longer thoroughly explain ALS pathogenesis. Exposure to various environmental factors, such as EMFs, solvents, heavy metals, and agricultural pesticides, has been hypothesised to contribute to ALS pathogenesis. Nevertheless, the contribution of several occupational factors to ALS has proven difficult to assess since there are no clear signs indicative of ALS development in ostensibly healthy individuals. Therefore, exposure to various environmental factors is based solely on patient recollection, and existing studies have provided controversial evidence that has not clarified the role of environmental factors in ALS (Figure 2).

### 4.1. EMFs

EMFs occur naturally and are ever present in people’s lives. However, environmental exposure has increased lately due to the rapid development of artificial EMF sources [66]. There are generally two EMF types: low frequency (LF-EMF) from power lines, household electrical appliances, and computers, and high frequency (HF-EMF) from radars, radios, mobile phones, and television broadcast towers [66]. Most studies have focused on the LF-EMF but provide conflicting results regarding LF-EMF-induced neurodegeneration. A Swedish meta-analysis found a potential positive association between LF-EMF exposure and ALS occurrence. However, they could not discriminate between isolated and occupational LF-EMF exposure [67]. An updated study by the same group found that LF-EMF occupational exposure carries an absolute risk for ALS [68]. A prospective study that followed patients for 17.3 years found a positive exposure-dependant association between occupational exposure to shallow frequency magnetic fields and ALS mortality [69]. An Italian study further supports these findings, confirming the predisposing effect of the proximity to overhead power lines on increasing ALS risk [70]. However, a Dutch study found no elevated risk of ALS development in people living near LF-EMF sources [71]. Therefore, these studies should be interpreted cautiously, as their correlations are based broadly on registry data, which is much more susceptible to bias [72,73]. Evidence for HF-EMFs is limited. Luna et al. [66] performed an epidemiological study that hypothesised a potential association between HF-EMF exposure from mobile communication antennas and ALS occurrence. However, further studies are needed to confirm or refute these findings.

The pathological implications of EMF exposure on the nervous system remain under investigation. Various neurological effects have been reported, such as disturbances in circadian rhythm, altered cognitive function, abnormal neuronal electrical activity, modified neurotransmitters release (e.g., glutamate-mediated excitotoxicity), elevated ROS production, and impaired blood-brain barrier permeability [66,74]. Both in vivo and in vitro studies have found that EMF-induced OS significantly influences different cellular processes, including gene dysregulation, abnormal protein aggregation, and neuroinflammation, which all have established roles in ALS [72,73,74,75]. Several studies have reported a substantial deterioration in antioxidant defence mechanisms in aged rats exposed to LF-EMF [76,77,78]. Moreover, exposure to high LF-EMF leads to neurological effects consequent not only to lipid peroxidation, but also to disturbances in several molecular processes, such as iron-related gene dysregulation in SOD1 mouse mutant models [78,79]. Other murine studies on mutant SOD1 have found no apparent EMF effect on ALS onset and survival. However, motor performances appeared worse after exposure compared to unexposed controls [76,80].

### 4.2. Solvents

Solvents are increasingly present in modern society due to their inclusion in different industrial and household products, such as paints, adhesives, and cleaning solutions. They can penetrate the blood-brain barrier through their lipophilic properties and cause various neurological disturbances, from cognitive impairment to motor deficits. Moreover, solvents accumulate within fatty tissues with recurrent exposure and continue to damage nerve function [81,82]. While solvents have gained increasing recognition as ALS inducers, findings on the relationship between solvent exposure and ALS pathogenesis have been contradictory. Koeman et al. [69] disproved any potential connection between ALS and aromatic and chlorinated solvents. However, a Swedish case-control study [83] found an inverse association between methylene chloride exposure, a common solvent with known carcinogenic effects in humans, and ALS risk in people younger than 65, contrasting with a previous study [84] that found no connection between them. Formaldehyde exposure, particularly among healthcare workers, has provided conflicting results. Some studies support its role in increasing ALS risk, particularly in males [83,85,86], while another study found little or no association [84]. However, a positive association between aromatic solvents and ALS was reported by Malek et al. [87]. Moreover, Andrew et al. [88] concluded that higher solvent exposure increased ALS risk in industrial employees, consistent with the findings of Malek et al. in a residential setting [89].

Volatile organic compounds (VOCs), particularly toluene and xylene, are most frequently implicated in impairing central nervous system (CNS) function. Their primary pathological mechanism is OS due to GSH depletion [81,90]. Studies have shown that GSH levels decrease as ALS progresses, supporting the hypothesis that VOC exposure might contribute to inducing latent ALS [81,91,92,93]. Organic solvents also deplete mitochondrial ATP, leading to abnormal functioning of ATP-dependent cellular processes and eventually apoptosis [81,94]. Constant VOC exposure is associated with increased neuronal excitation. Given the already recognised involvement of motor neuron hyperexcitability in ALS pathogenesis, the latter mechanism supports the potential role of VOCs in ALS progression [81,95]. Furthermore, toluene exposure has been shown to intervene in axonal transport by reducing levels of microtubule-associated protein 2 (MAP2), leading to the loss of anterior horn neurons in ALS patients [81,96,97,98].

### 4.3. Heavy Metals

Heavy metals naturally occur in the Earth’s crust. Some, such as iron, zinc, and copper, have adverse physiological effects on different metabolic processes. Others adversely affect body homeostasis, even in small amounts, leading to various severe disorders, including neurodegeneration [99]. The general population can be exposed to heavy metals in various situations. The most frequent cause is occupational environment-related, such as plumbing, welding, and metal manufacturing. Exposure via contaminated food, water, medication, dental fillings, and air pollution has also been reported [100,101]. While heavy metal exposure has been hypothesised to contribute to ALS development, studies required to confirm this relationship are difficult to perform. Most studies have based their conclusions on blood-borne heavy metal levels measured after ALS diagnosis, which might be caused by ALS progression [100]. For instance, a Swedish meta-analysis found an approximately 50% increased risk of ALS development in people exposed to heavy metals based on higher blood-borne lead levels in ALS patients compared to matched controls.

Nevertheless, this study’s authors have identified some publication bias in the included studies [67]. The following year, they assessed the relationship between lead and non-lead (e.g., a mixture of other metals) exposure and the occurrence of neurodegenerative disorders. Therefore, lead exposure has proven to be a particular risk factor in ALS development, while non-lead exposure has not [68]. Filippini et al. [70] also found increased levels of lead and other heavy metals, such as selenium, in blood and CSF from ALS patients compared with healthy controls, concluding a positive association between heavy metal exposure and increased ALS risk. Andrew et al. [102] found a positive association between increased mercury fingernail levels and ALS diagnosis. Similarly, other studies have reported elevated mercury levels in ALS patients` hair [103,104,105]. In addition, Koeman et al. [69] found a negligible influence of heavy metal exposure on ALS mortality. However, the prevalence of metal exposure among subjects included in the study was relatively low. Dickerson et al. [106] reported a non-significant association between job exposure to chromium, iron, and nickel and the ALS risk. Kullmann et al. [101] concluded that mercury exposure from seafood and dental amalgams is insufficient to induce ALS. However, mercury already within the motor neurons might trigger sporadic ALS form when combined with a genetic or epigenetic predisposition to ALS. To clarify the effects of heavy metal exposure on ALS pathogenesis, Peters et al. [100] performed the first prospective study assessing prediagnostic metal blood levels in ALS patients compared to matched controls. Erythrocyte heavy metal levels were used to quantify exposure, considering two critical facts: metals tend to bind to red blood cells, and the erythrocyte lifespan of approximately three months makes them a suitable indicator of ongoing exposure. This study concluded that lead and cadmium might elevate ALS risk, while zinc decreases ALS risk.

Various mechanisms have been proposed for the influence of heavy metals on ALS pathogenesis. Wang et al. [107] performed a meta-analysis and systematic review that provided evidence of heavy metal exposure associated with elevated ALS risk. The most extensively studied element and the primary cause was lead, followed closely by mercury, although studied to a much lesser extent. Lead accumulates in the blood, mineral, and soft tissues, particularly the kidney, liver, and brain, where it can permeate the blood-brain barrier and accumulate within neurons and glial cells [108]. Neurological malfunctions are caused by demyelination and impaired neurotransmission due to lead’s mimicry of divalent metals, such as calcium and zinc, altering homeostasis and inducing OS [108]. High blood and CSF lead levels have been reported in ALS patients [99,109]. Nevertheless, it should be noted that lead contained within the bones is released into the blood during bone remodelling, which occurs faster in ALS patients, particularly those with severe ALS, elevating blood lead levels [99]. Wang et al. [107], found that ALS induced by lead exposure has a longer survival period potentially reflecting increased antioxidant and vascular endothelial growth factor (VEGF) production by glial cells and motor neurons, stimulated by prior lead exposure [108,110,111,112].

Mercury has recognised neurotoxic effects, particularly on the axons of neuron cells whose damage contributes to ALS pathogenesis [107]. Furthermore, Ash et al. [113] found that mercury and lead are involved in TARDBP malfunction and structural abnormalities, which are recognised molecular hallmarks in ALS pathogenesis. TARDBP is also responsible for ROS production, autoimmunity enhancement, impaired nucleic acid and protein synthesis, and dysfunction in various organelles, such as mitochondria and endoplasmic reticula [101]. Chromium and nickel have adverse effects on oxidative equilibrium and contribute to carcinogenesis. Iron is indispensable for normal nervous system physiology. However, high iron levels lead to neurodegeneration in ALS patients. Iron chelation has proven therapeutic in ALS murine models [99,106,114]. Zinc is essential for the correct conformation and function of various enzymes involved in human cell metabolism. Zinc deficiency has been associated with SOD1 dissociation and protein aggregation, leading to oxidation-induced motor neuron cell death in ALS patients. Therefore, ALS patients with zinc deficiencies have a worse prognosis [99,115].

Copper is also an essential cofactor for the proper nervous system functioning as a coregulator of SOD1 stability and activity with zinc and a regulator of oxidative balance [115,116]. Mutant SOD1 involvement in ALS pathophysiology is closely associated with copper deficiency. Interestingly, abnormal TARDBP aggregation has been associated with copper homeostasis. Proper copper levels protect against TARDBP accumulation, while mutant TARDBP overexpression is associated with abnormally high copper levels in the spinal cord [117]. Elevated copper levels lead to free radical accumulation and potentially increase ALS risk [99]. Moreover, Patti et al. [115] found higher copper and manganese levels in ALS patients with bulbar onset compared to those with spinal onset. Since bulbar ALS has the worst prognosis, this association might be an important predictor of ALS progression [115]. Consequently, physicians face an important dilemma concerning whether neurotoxicity occurs due to inadequate copper levels in various essential enzymes, such as SOD1 or excessive copper accumulation [117].

Other metals, such as selenium, cadmium, aluminium, and manganese, have also been reported to induce ALS. Selenium has widely recognised antioxidant and antitumoral properties. Both selenium scarcity and excess are associated with neurological disturbances. However, its role in ALS development remains controversial. Some studies have suggested that selenium has protective properties in ALS patients due to its antioxidant function. In contrast, low levels increase ROS production, resulting in cellular death and neurological damage [99,118,119]. Nevertheless, studies on animal models have found spinal motor neuron degeneration caused by selenium exposure via OS, reduced cholinergic signalling, and SOD1 protein aggregation within motoneurons leading to cell death [115,120].

Cadmium has no recognised biological functions and is harmful even at minute concentrations. While its precise mechanism in ALS development remains under investigation, multiple hypotheses have recently emerged. One proposes that SOD1 inactivation is caused by cadmium replacing zinc in its structure, forming metallothionein within motor neurons and astrocytes in ALS patients [99,121,122]. Cadmium is also involved in motor neuron apoptosis, which appears due to either ionic disequilibrium (elevated cadmium levels increase intracellular calcium levels) or disturbances of different proteins involved in cellular death, such as VDAC1 and protein disulfide isomerase (PDI). Moreover, cadmium promotes OS [122].

While aluminium has pro-oxidant properties, it promotes ROS production, weakening the antioxidant defence. Increased aluminium levels within nervous system components in ALS patients support its involvement in ALS pathogenesis, which remains to be fully understood [99,123]. Manganese is another heavy metal whose presence in the anterior horn cells, which are the first to degenerate in ALS, has been shown [124]. Moreover, it can cross the blood-brain barrier and impair neuronal cell metabolism, particularly ATP production, leading to a pro-oxidative status [125].

### 4.4. Agricultural Pesticides

Pesticides are another class of potential environmental factors influencing ALS, whose exposure can be either occupational, particularly among farmers, or non-occupational through ingestion, inhalation, and direct skin contact. To date, results are somewhat contradictory and contentious due to selection and confounding biases [126]. Several Swedish meta-analyses based on systemic literature reviews assessed the influence of occupational risk factors on ALS development from an epidemiological perspective. They concluded that ALS is most likely to occur in those exposed to chemicals, particularly agricultural pesticides [67], which is supported by a more recent meta-analysis [68]. An Italian population-based case-control study showed an increased risk of ALS development in agricultural workers exposed to different pesticides.

Furthermore, ALS risk appears to be related to exposure duration. According to one study, fungicides are the most significant threat [70]. Furthermore, an outbreak of organophosphate-induced delayed polyneuropathy (OPIDP) due to soil contamination in the 1940s was described on an Italian farm. OPIDP manifested as progressive, distal to proximal, motor nerve damage, with no sensory involvement and symptomatology similar to current ALS criteria [107,127]. Moreover, Burns et al. [128] found three times greater mortality in ALS patients working in a pesticide plant. Other studies have reported contradictory results, with a non-significant negative association between ALS mortality and exposure to pesticides, herbicides, and insecticides [69]. Vinceti et al. [126] reported little or no increased risk for ALS in people living near agricultural land where they are heavily exposed to pesticides. Furthermore, no significant association was found between ALS risk and CSF levels of various pesticides, including organochlorine, polychlorinated biphenyls, and polycyclic aromatic hydrocarbons [129].

Emerging evidence supports a positive association between ALD and pesticides, particularly OPs. The underlying mechanisms involve OS, neuro-inflammation, cholinergic deficits, and epigenetic disturbances. Genetic studies found PON1, an enzyme involved in organophosphate pesticides detoxification, to be more frequently mutated in ALS patients than in healthy individuals [62,107]. Furthermore, OPs irreversibly block cholinesterase and lead to excessive acetylcholine accumulation in the CNS, altering normal cholinergic transmission, which is essential for numerous neurological processes. Consequently, individuals chronically exposed to OPs have cholinergic neuronal dysfunction and can develop a broad range of neurological symptoms, from motor deficits to cognitive impairment [130,131]. Repeated cholinergic stimulations eventually exhaust mitochondrial ATP synthesis and create a suitable environment for ROS overproduction, a well-known mechanism in ALS pathogenesis [130,132].

## 5. ALS Therapies Effects on OS Pathways

The most horrific feature of ALS is its incurable nature regardless of how early the diagnosis is determined and the treatment is initiated. In spite of hundreds of clinical trials performed since its discovery, only two therapeutic alternatives are currently approved by the international forums for ALS management, riluzole and edaravone [133]. Riluzole has been proven to increase the overall survival up to three months [134,135], while edaravone has been associated with a better preservation of motor functions of ALS patients as long as it is administered in the early phase of the disease [136,137,138,139]. The precise mechanisms of action of these drugs are not completely recognized, yet it is already demonstrated their involvement in the reduction of pro-oxidative status in the motor neurons.

Riluzole essentially protects against glutamate excitotoxicity, one of the main molecular pathways involved in ALS pathogenesis. By blocking the presynaptic secretion of glutamate, it prevents the excessive accumulation of calcium inside the cytoplasm of motor neurons and glial cells, which would otherwise lead to mitochondrial damage and, eventually, to cell death by increased ROS production [133].

Edaravone exhibits its neuroprotective effects by scavenging ROS and inhibiting apoptosis [140]. Its great permeability for cell membrane and blood-brain barrier due to its lipophilic structure allows Edaravone to directly protect brain cells. One of the main mechanisms is upregulation of peroxiredoxin-2, a molecule involved in inhibition of cellular death by stimulating apoptosis signal regulating kinase (ASK1) signaling cascade [141]. Edavarone also serves as a scavenging system for free radicals and, thus, prevents the OS propagation [142].

The confirmation of OS and mitochondrial dysfunction role in ALS pathogenesis has opened a novel research field with emphasis on drugs interfering with redox processes. Multiple studies have been performed regarding antioxidant features of various molecules, such as creatine, vitamin C, and vitamin E. However, further clinical trials have infirmed their utility as ALS curative treatment on a long-term basis [143].

Another research focus was on mitochondria-targeting drugs, such as mitoquinone (MitoQ), which is naturally occurring in mitochondria ubiquinone conjugated with triphenylphosphonium (TPP^+^). MitoQ possesses increased permeability for all biologic membranes, blood-brain barrier included, and high affinity for mitochondria, features that would promise remarkable neuroprotective properties by decreasing oxidative damage and maintaining normal mitochondrial function [143]. Studies performed on SOD^G93A^ murine models have shown extended preservation of motor and mitochondrial functions and a longer survival after oral treatment with MitoQ [144]. Nonetheless, similar results have failed to appear on ALS human patients to date [145]. Szeto-Schiller peptides are molecules acting directly on the mitochondrial site where ROS are generated. Murine studies focused on their neuroprotective effects have shown favourable results, yet clinical trials are still lacking [143].

A remarkable initiative refers to molecules intended to sharpen the antioxidant response by upregulating various genes containing the antioxidant response element (ARE) [143]. For instance, sulforaphane irreversibly binds to Keap1 and, consequently, activates Nrf2/ARE antioxidant pathway [146]. However, the results regarding its role in ALS treatment are inconclusive [147]. A more specific Keap1 inhibitor is p62-mediated mitophagy inducer (PIM). It promotes a process of quality control among all mitochondria without interfering with their energetic function and induces autophagy only in damaged organelle [143,148]. PIM remains under investigation as a potential ALS therapeutic molecule, since the inhibition of selective autophagy is a recognized mechanism in ALS pathogenesis [149].

Another promising pharmacological molecule in ALS treatment is CuATSM (diacetylbis(N(4)-methylthiosemicarbazonato)copper(II)). It can easily penetrate the blood-brain barrier and selectively and safely provide copper to cells with altered mitochondria [150]. Williams et al. [150] have demonstrated an extended survival in SOD^G93A^ mice treated with CuATSM, whereas Lum et al. [151] have additionally observed an important delay in ALS onset without treatment-related side effects. A multicentre, randomized, double-blind, placebo controlled clinical trial is currently ongoing in Australia evaluating the CuATSM treatment in ALS human patients in terms of tolerability and efficacy [151].

## 6. Conclusions

Exposure to environmental factors or possessing mutations in ALS-associated genes may cause mitochondrial dysfunction. These mechanisms share common linkages and are interconnected. Neuroprotective strategies must address multiple pathological pathways, such as mitochondrial dysfunction, and avoid exposure to modifiable environmental factors related to ALS. Understanding the multifaceted relationship between genetic and environmental risk factors, with OS as a common denominator underlying neurodegeneration, will help to develop neuroprotective strategies.

ALS is a neurodegenerative disorder with devastating disabilities that rapidly progress to death. As its etiology is still unclear, effective curative treatment is not yet available. Therefore, massive resources have been invested in research in order to better understand the underlying mechanisms for ALS occurrence. Currently, it is considered a multi-factorial disease, triggered by complex interactions among different cellular and molecular processes. Oxidative stress is one major contributor to ALS pathogenesis. It appears as a result of increased intracellular levels of highly reactive free radicals combined with defective compensatory antioxidant systems. More frequently, OS-inducing disequilibrium is due to mutations occurring in different genes already known to play an important role in ALS etiology. Less commonly, OS might appear as a consequence of external environmental factors, whose specific involvement, however, has not been yet fully clarified. Further research is crucial for elucidating the precise mechanisms that underlies OS production in ALS in order to develop a targeted, individualized treatment.

## Figures and Tables

**Figure 1 ijms-23-09339-f001:**
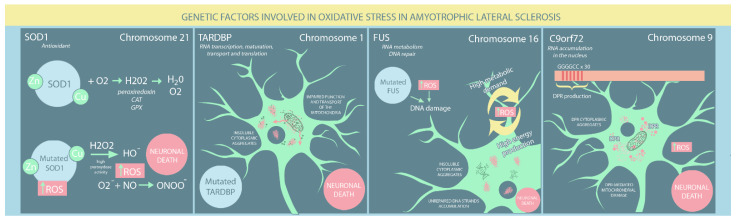
Genetic risk factors involved in oxidative stress in ALS patients. (SOD: superoxide dismutase; ROS: reactive oxygen species; DPR: dipeptide repeat proteins).

**Figure 2 ijms-23-09339-f002:**
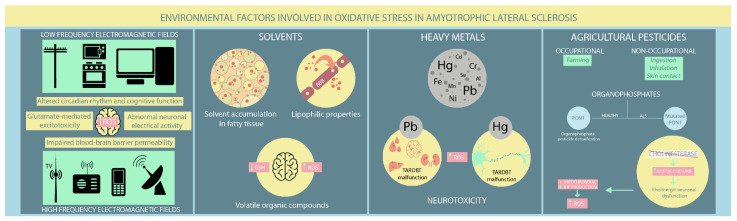
Environmental risk factors involved in oxidative stress in ALS patients. (ROS: reactive oxygen species; Hg: mercury; Pb: lead; Cd: cadmium; Cr: chromium; Se: selenium; Fe: iron; Mn: manganese; Ni: nickel).

## Data Availability

Not applicable.

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
