# Peer review of "Oxidative Stress in Amyotrophic Lateral Sclerosis: Synergy of Genetic and Environmental Factors"

_ijms, 2022, doi:10.3390/ijms23169339_

Round 1

Reviewer 1 Report

In this paper, Motataianu et al. aimed to review the oxidative stress in amyotrophic lateral sclerosis from the perspective of the genetic and environmental factors. It's a fascinating topic with many therapeutic attempts in recent years. The review summarizes the current status in this area, however the overall study is still relatively superficial. If the author can have a more in-depth consideration about the future direction of research in this area, it would be more meaningful to readers.

There are some considerations that I think need to be addressed to improve this study.

1.     Although genetic and environmental factors are widely recognized risk factors in ALS, it is well known that more patients are sporadic and have no environmental exposures. It’s better to review the oxidative stress in general ALS first, and then to emphasize the roles of the above risk factors.

2.     In recent years, the free radical scavenger edaravone has been approved for the treatment of ALS. In addition, there are many clinical trials targeting mitochondrial function, such as MitoQ, and some other free radical scavengers are developed. Although consistent conclusions about the results of these trials are still lacking, the mechanisms related with oxidative stress involved in these treatments of interest should be reviewed in this article. It will be appreciated to summarize the potential therapeutic targets with respect of oxidative stress.

3.     The interaction between oxidative stress and pathological proteins, such as TDP-43,   should be emphasized. On the one hand, pathological proteins increase free radical production and impaired mitochondrial function, which have been described in detail in the manuscript. On the other hand, the respiration-related ROS can enhance the toxicity of TDP-43 (for example: Park et al. PMID: 30905713; Gautam et al. PMID: 30450515), which was not mentioned.

4.     Other comments:

Line 27-28: Not all the familial ALS have the AD pattern.

Line 31-32: Antioxidant intake maybe a protective factor for ALS, which was not proper to be listed here.

Line 100-101: SOD1 mutation accounts for about 20% of familial ALS.

The figures have very low resolution, especially the text is sometimes not readable.

Author Response

Thank you very much for your thorough review! We deeply appreciate your effort to help us increase our paper's scientific value. By adding the final chapter focusing on how current ALS treatment and other under investigation therapeutic molecules might interfere and favourably modify the existing pro-oxidative status in ALS, we consider that our review become more meaningful to the readers.

Please see below the modifications we made in order to comply with your requirements. We also submitted the new version of our review in the attachment.

1) “Although genetic and environmental factors are widely recognized risk factors in ALS, it is well known that more patients are sporadic and have no environmental exposures. It’s better to review the oxidative stress in general ALS first, and then to emphasize the roles of the above risk factors.”

Answer: We shortly reviewed the mitochondrial dysfunction and associated oxidative stress in ALS (please see chapter 2 of our article). We afterwards focused on the influence of genetic and environmental factors on oxidative stress in ALS (please see chapters 3 and 4). If you do not consider chapter 2 to be comprehensible enough, please let us know and we will revise its content.

2) “In recent years, the free radical scavenger edaravone has been approved for the treatment of ALS. In addition, there are many clinical trials targeting mitochondrial function, such as MitoQ, and some other free radical scavengers are developed. Although consistent conclusions about the results of these trials are still lacking, the mechanisms related with oxidative stress involved in these treatments of interest should be reviewed in this article. It will be appreciated to summarize the potential therapeutic targets with respect of oxidative stress.”

Answer: Please see the newly added chapter 4 entitled “ALS therapies effects on OS pathways” at the end of our article.

3) “The interaction between oxidative stress and pathological proteins, such as TDP-43,   should be emphasized. On the one hand, pathological proteins increase free radical production and impaired mitochondrial function, which have been described in detail in the manuscript. On the other hand, the respiration-related ROS can enhance the toxicity of TDP-43 (for example: Park et al. PMID: 30905713; Gautam et al. PMID: 30450515), which was not mentioned.”

Answer: Please see lines 192-209 and references 41,42.

4) “Line 27-28: Not all the familial ALS have the AD pattern.”

Answer: Please see lines 28-30.

5) “Line 31-32: Antioxidant intake maybe a protective factor for ALS, which was not proper to be listed here.”

Answer: Please see lines 32-33.

6) “Line 100-101: SOD1 mutation accounts for about 20% of familial ALS.”

Answer: Please see lines 104-105.

7) “The figures have very low resolution, especially the text is sometimes not readable.”

Answer: We modified the article’s format and put the figures at the end at a higher resolution and a bigger size.

Reviewer 2 Report

A very nice manuscript that will provide a useful review on how oxidative stress might be involved in ALS along with some information on the most studied pro-oxidants. I appreciated the fact that the authors did not make a strong case for the involvement of a particular agent in the pathogenesis of ALS, but rather they presented all sides of a story and remained objective in their descriptions.

I would have liked to see a small section on how existing approved therapeutics for ALS (riluzole, edaravone) may effect the OS pathways mentioned in the review.  Also, some speculation on why several antioxidant therapies have failed to produced a favorable outcome in the clinic and why some others, which are under investigation, may yield benefit (e.g. CuATSM).

Some small comments and edits:

line 15: It's not just lately that OS has been demonstrated to play a crucial role in ALS pathogenesis. We've known that for a while. The word "lately" should be removed and replaced with the word "also" as in "OS has also been demonstrated...."

line 26: grammar. It should be: The etiology of ALS is not yet...

line 101: i believe SOD1 mutations account for 2% of all ALS cases. Not 20%. And around 10% within the familial cases.

line 267: the statement on ALS occurring at younger ages requires a reference.

Author Response

Thank you very much for your thorough review! We deeply appreciate your effort to help us increase our paper's scientific value.

Please see below the modifications we made in order to comply with your requirements. We also submitted the new version of our review in the attachment below.

1) “I would have liked to see a small section on how existing approved therapeutics for ALS (riluzole, edaravone) may effect the OS pathways mentioned in the review.  Also, some speculation on why several antioxidant therapies have failed to produced a favorable outcome in the clinic and why some others, which are under investigation, may yield benefit (e.g. CuATSM).”

Answer: We added a new chapter entitled `ALS therapies effects on OS pathways` at the end of our article, in which we resumed the influence of various therapeutic molecules on changing the existing pro-oxidative status in ALS.

2) “line 15: It's not just lately that OS has been demonstrated to play a crucial role in ALS pathogenesis. We've known that for a while. The word "lately" should be removed and replaced with the word "also" as in "OS has also been demonstrated...."

Answer: Please see line 15, we replaced “lately” with “notably”.

3) “line 26: grammar. It should be: The etiology of ALS is not yet...”

Answer: Please see line 27.

4) “line 101: i believe SOD1 mutations account for 2% of all ALS cases. Not 20%. And around 10% within the familial cases.”

Answer: Please see lines 104-105.

5) “line 267: the statement on ALS occurring at younger ages requires a reference.”

Answer: Please see line 273.
